# Halting in Random Walk Kernels

**Mahito Sugiyama**
ISIR, Osaka University, Japan
JST, PRESTO
mahito@ar.sanken.osaka-u.ac.jp

**Karsten M. Borgwardt**
D-BSSE, ETH Zürich
Basel, Switzerland
karsten.borgwardt@bsse.ethz.ch

## Abstract

Random walk kernels measure graph similarity by counting matching walks in two graphs. In their most popular form of geometric random walk kernels, longer walks of length $k$ are downweighted by a factor of $\lambda^k$ ($\lambda < 1$) to ensure convergence of the corresponding geometric series. We know from the field of link prediction that this downweighting often leads to a phenomenon referred to as *halting*: Longer walks are downweighted so much that the similarity score is completely dominated by the comparison of walks of length 1. This is a naïve kernel between edges and vertices. We theoretically show that halting may occur in geometric random walk kernels. We also empirically quantify its impact in simulated datasets and popular graph classification benchmark datasets. Our findings promise to be instrumental in future graph kernel development and applications of random walk kernels.

## 1 Introduction

Over the last decade, graph kernels have become a popular approach to graph comparison [4, 5, 7, 9, 12, 13, 14], which is at the heart of many machine learning applications in bioinformatics, imaging, and social-network analysis. The first and best-studied instance of this family of kernels are *random walk kernels*, which count matching walks in two graphs [5, 7] to quantify their similarity. In particular, the geometric random walk kernel [5] is often used in applications as a baseline comparison method on graph benchmark datasets when developing new graph kernels. These geometric random walk kernels assign a weight $\lambda^k$ to walks of length $k$, where $\lambda < 1$ is set to be small enough to ensure convergence of the corresponding geometric series.

Related similarity measures have also been employed in link prediction [6, 10] as a similarity score between vertices [8]. However, there is one caveat regarding these approaches. Walk-based similarity scores with exponentially decaying weights tend to suffer from a problem referred to as *halting* [1]. They may downweight walks of lengths 2 and more, so much so that the similarity score is ultimately completely dominated by walks of length 1. In other words, they are almost identical to a simple comparison of edges and vertices, which ignores any topological information in the graph beyond single edges. Such a simple similarity measure could be computed more efficiently outside the random walk framework. Therefore, halting may affect both the expressivity and efficiency of these similarity scores.

Halting has been conjectured to occur in random walk kernels [1], but its existence in graph kernels has never been theoretically proven or empirically demonstrated. Our goal in this study is to answer the open question if and when halting occurs in random walk graph kernels.

We theoretically show that halting may occur in graph kernels and that its extent depends on properties of the graphs being compared (Section 2). We empirically demonstrate in which simulated datasets and popular graph classification benchmark datasets halting is a concern (Section 3). We conclude by summarizing when halting occurs in practice and how it can be avoided (Section 4).

We believe that our findings will be instrumental in future applications of random walk kernels and the development of novel graph kernels.

## 2 Theoretical Analysis of Halting

We theoretically analyze the phenomenon of halting in random walk graph kernels. First, we review the definition of graph kernels in Section 2.1. We then present our key theoretical result regarding halting in Section 2.2 and clarify the connection to linear kernels on vertex and edge label histograms in Section 2.3.

### 2.1 Random Walk Kernels

Let $G = (V, E, \varphi)$ be a labeled graph, where $V$ is the vertex set, $E$ is the edge set, and $\varphi$ is a mapping $\varphi : V \cup E \to \Sigma$ with the range $\Sigma$ of vertex and edge labels. For an edge $(u, v) \in E$, we identify $(u, v)$ and $(v, u)$ if $G$ is undirected. The degree of a vertex $v \in V$ is denoted by $d(v)$.

The direct (tensor) product $G_\times = (V_\times, E_\times, \varphi_\times)$ of two graphs $G = (V, E, \varphi)$ and $G' = (V', E', \varphi')$ is defined as follows [1, 5, 14]:

$$V_\times = \{ (v, v') \in V \times V' \mid \varphi(v) = \varphi'(v') \},$$
$$E_\times = \{ ((u, u'), (v, v')) \in V_\times \times V_\times \mid (u, v) \in E,\ (u', v') \in E',\ \text{and } \varphi(u, v) = \varphi'(u', v') \},$$

and all labels are inherited, or $\varphi_\times((v, v')) = \varphi(v) = \varphi'(v')$ and $\varphi_\times((u, u'), (v, v')) = \varphi(u, v) = \varphi'(u', v')$. We denote by $A_\times$ the adjacency matrix of $G_\times$ and denote by $\delta_\times$ and $\Delta_\times$ the minimum and maximum degrees of $G_\times$, respectively.

To measure the similarity between graphs $G$ and $G'$, random walk kernels count all pairs of matching walks on $G$ and $G'$ [2, 5, 7, 11]. If we assume a uniform distribution for the starting and stopping probabilities over the vertices of $G$ and $G'$, the number of matching walks is obtained through the adjacency matrix $A_\times$ of the product graph $G_\times$ [14]. For each $k \in \mathbb{N}$, the $k$-step random walk kernel between two graphs $G$ and $G'$ is defined as:

$$K_\times^k(G, G') = \sum_{i,j=1}^{|V_\times|} \left[ \sum_{l=0}^{k} \lambda_l A_\times^l \right]_{ij}$$

with a sequence of positive, real-valued weights $\lambda_0, \lambda_1, \lambda_2, \ldots, \lambda_k$ assuming that $A_\times^0 = \mathbf{I}$, the identity matrix. Its limit $K_\times^\infty(G, G')$ is simply called the *random walk kernel*.

Interestingly, $K_\times^\infty$ can be directly computed if weights are the geometric series, or $\lambda_l = \lambda^l$, resulting in the geometric random walk kernel:

$$K_{\mathrm{GR}}(G, G') = \sum_{i,j=1}^{|V_\times|} \left[ \sum_{l=0}^{\infty} \lambda^l A_\times^l \right]_{ij} = \sum_{i,j=1}^{|V_\times|} \left[ (\mathbf{I} - \lambda A_\times)^{-1} \right]_{ij}.$$

In the above equation, let $(\mathbf{I} - \lambda A_\times)\boldsymbol{x} = \mathbf{0}$ for some value of $\boldsymbol{x}$. Then, $\lambda A_\times \boldsymbol{x} = \boldsymbol{x}$ and $(\lambda A_\times)^l \boldsymbol{x} = \boldsymbol{x}$ for any $l \in \mathbb{N}$. If $(\lambda A_\times)^l$ converges to 0 as $l \to \infty$, $(\mathbf{I} - \lambda A_\times)$ is invertible since $\boldsymbol{x}$ becomes $\mathbf{0}$. Therefore, $(\mathbf{I} - \lambda A_\times)^{-1} = \sum_{l=0}^{\infty} \lambda^l A_\times^l$ from the equation $(\mathbf{I} - \lambda A_\times)(\mathbf{I} + \lambda A_\times + \lambda^2 A_\times^2 + \ldots) = \mathbf{I}$ [5]. It is well-known that the geometric series of matrices, often called the Neumann series, $\mathbf{I} + \lambda A_\times + (\lambda A_\times)^2 + \cdots$ converges only if the maximum eigenvalue of $A_\times$, denoted by $\mu_{\times,\max}$, is strictly smaller than $1/\lambda$. Therefore, the geometric random walk kernel $K_{\mathrm{GR}}$ is well-defined only if $\lambda < 1/\mu_{\times,\max}$.

There is a relationship for the minimum and maximum degrees $\delta_\times$ and $\Delta_\times$ of $G_\times$ [3]: $\delta_\times \leq \bar{d}_\times \leq \mu_{\times,\max} \leq \Delta_\times$, where $\bar{d}_\times$ is the average of the vertex degrees of $G_\times$, or $\bar{d}_\times = (1/|V_\times|) \sum_{v \in V_\times} d(v)$. In practice, it is sufficient to set the parameter $\lambda < 1/\Delta_\times$.

In the inductive learning setting, since we do not know *a priori* target graphs that a learner will receive in the future, $\lambda$ should be small enough so $\lambda < 1/\mu_{\times,\max}$ for any pair of unseen graphs. Otherwise, we need to re-compute the full kernel matrix and re-train the learner. In the transductive

setting, we are given a collection $\mathcal{G}$ of graphs beforehand. We can explicitly compute the upper bound of $\lambda$, which is $(\max_{G,G'\in\mathcal{G}}\mu_{\times,\max})^{-1}$ with the maximum of the maximum eigenvalues over all pairs of graphs $G, G' \in \mathcal{G}$.

## 2.2 Halting

The geometric random walk kernel $K_{\mathrm{GR}}$ is one of the most popular graph kernels, as it can take walks of any length into account [5, 14]. However, the fact that it weights walks of length $k$ by the $k$th power of $\lambda$, together with the condition that $\lambda < (\mu_{\times,\max})^{-1} < 1$, immediately tells us that the contribution of longer walks is significantly lowered in $K_{\mathrm{GR}}$. If the contribution of walks of length 2 and more to the kernel value is even completely dominated by the contribution of walks of length 1, we would speak of halting. It is as if the random walks halt after one step.

Here, we analyze under which conditions this halting phenomenon may occur in geometric random walk kernels. We obtain the following key theoretical statement by comparing $K_{\mathrm{GR}}$ to the one-step random walk kernel $K_{\times}^1$.

**Theorem 1** *Let $\lambda_0 = 1$ and $\lambda_1 = \lambda$ in the random walk kernel. For a pair of graphs $G$ and $G'$,*
$$K_{\times}^1(G, G') \le K_{\mathrm{GR}}(G, G') \le K_{\times}^1(G, G') + \varepsilon,$$
*where*
$$\varepsilon = |V_{\times}|\frac{(\lambda\Delta_{\times})^2}{1 - \lambda\Delta_{\times}},$$
*and $\varepsilon$ monotonically converges to $0$ as $\lambda \to 0$.*

*Proof.* Let $d(v)$ be the degree of a vertex $v$ in $G_{\times}$ and $N(v)$ be the set of neighboring vertices of $v$, that is, $N(v) = \{u \in V_{\times} \mid (u, v) \in E_{\times}\}$. Since $A_{\times}$ is the adjacency matrix of $G_{\times}$, the following relationships hold:

$$\sum_{i,j=1}^{|V_{\times}|}[A_{\times}]_{ij} = \sum_{v\in V_{\times}}d(v) \le |V_{\times}|\Delta_{\times}, \quad \sum_{i,j=1}^{|V_{\times}|}[A_{\times}^2]_{ij} = \sum_{v\in V_{\times}}\sum_{v'\in N(v)}d(v') \le |V_{\times}|\Delta_{\times}^2,$$

$$\sum_{i,j=1}^{|V_{\times}|}[A_{\times}^3]_{ij} = \sum_{v\in V_{\times}}\sum_{v'\in N(v)}\sum_{v''\in N(v')}d(v'') \le |V_{\times}|\Delta_{\times}^3, \ldots, \quad \sum_{i,j=1}^{|V_{\times}|}[A_{\times}^n]_{ij} \le |V_{\times}|\Delta_{\times}^n.$$

From the assumption that $\lambda\Delta_{\times} < 1$, we have

$$K_{\mathrm{GR}}(G, G') = \sum_{i,j=1}^{|V_{\times}|}[\mathbf{I} + \lambda A_{\times} + \lambda^2 A_{\times}^2 + \ldots]_{ij} = K_{\times}^1(G, G') + \sum_{i,j=1}^{|V_{\times}|}[\lambda^2 A_{\times}^2 + \lambda^3 A_{\times}^3 + \ldots]_{ij}$$

$$\le K_{\times}^1(G, G') + |V_{\times}|\lambda^2\Delta_{\times}^2(1 + \lambda\Delta_{\times} + \lambda^2\Delta_{\times}^2 + \ldots) = K_{\times}^1(G, G') + \varepsilon.$$

It is clear that $\varepsilon$ monotonically goes to $0$ when $\lambda \to 0$. $\blacksquare$

Moreover, we can normalize $\varepsilon$ by dividing $K_{\mathrm{GR}}(G, G')$ by $K_{\times}^1(G, G')$.

**Corollary 1** *Let $\lambda_0 = 1$ and $\lambda_1 = \lambda$ in the random walk kernel. For a pair of graphs $G$ and $G'$,*
$$1 \le \frac{K_{\mathrm{GR}}(G, G')}{K_{\times}^1(G, G')} \le 1 + \varepsilon',$$
*where*
$$\varepsilon' = \frac{(\lambda\Delta_{\times})^2}{(1 - \lambda\Delta_{\times})(1 + \lambda\bar{d}_{\times})}$$
*and $\bar{d}_{\times}$ is the average of vertex degrees of $G_{\times}$.*

*Proof.* Since we have
$$K_{\times}^1(G, G') = |V_{\times}| + \lambda\sum_{v\in V_{\times}}d(v) = |V_{\times}|(1 + \lambda\bar{d}_{\times}),$$
it follows that $\varepsilon/K_{\times}^1(G, G') = \varepsilon'$. $\blacksquare$

Theorem 1 can be easily generalized to any $k$-step random walk kernel $K_{\times}^k$.

**Corollary 2** *Let $\varepsilon(k) = |V_\times|(\lambda\Delta_\times)^k/(1 - \lambda\Delta_\times)$. For a pair of graphs $G$ and $G'$, we have*
$$K_\times^k(G, G') \le K_{\mathrm{GR}}(G, G') \le K_\times^k(G, G') + \varepsilon(k+1).$$

Our results imply that, in the geometric random walk kernel $K_{\mathrm{GR}}$, the contribution of walks of length longer than 2 diminishes for very small choices of $\lambda$. This can easily happen in real-world graph data, as $\lambda$ is upper-bounded by the inverse of the maximum degree of the product graph.

## 2.3 Relationships to Linear Kernels on Label Histograms

Next, we clarify the relationship between $K_{\mathrm{GR}}$ and basic linear kernels on vertex and edge label histograms. We show that halting $K_{\mathrm{GR}}$ leads to the convergence of it to such linear kernels.

Given a pair of graphs $G$ and $G'$, let us introduce two linear kernels on vertex and edge histograms. Assume that the range of labels $\Sigma = \{1, 2, \ldots, s\}$ without loss of generality. The vertex label histogram of a graph $G = (V, E, \varphi)$ is a vector $\boldsymbol{f} = (f_1, f_2, \ldots, f_s)$, such that $f_i = |\{v \in V \mid \varphi(v) = i\}|$ for each $i \in \Sigma$. Let $\boldsymbol{f}$ and $\boldsymbol{f}'$ be the vertex label histograms of graphs $G$ and $G'$, respectively. The vertex label histogram kernel $K_{\mathrm{VH}}(G, G')$ is then defined as the linear kernel between $\boldsymbol{f}$ and $\boldsymbol{f}'$:
$$K_{\mathrm{VH}}(G, G') = \langle \boldsymbol{f}, \boldsymbol{f}' \rangle = \sum_{i=1}^s f_i f_i'.$$

Similarly, the edge label histogram is a vector $\boldsymbol{g} = (g_1, g_2, \ldots, g_s)$, such that $g_i = |\{(u, v) \in E \mid \varphi(u, v) = i\}|$ for each $i \in \Sigma$. The edge label histogram kernel $K_{\mathrm{EH}}(G, G')$ is defined as the linear kernel between $\boldsymbol{g}$ and $\boldsymbol{g}'$, for respective histograms:
$$K_{\mathrm{EH}}(G, G') = \langle \boldsymbol{g}, \boldsymbol{g}' \rangle = \sum_{i=1}^s g_i g_i'.$$

Finally, we introduce the vertex-edge label histogram. Let $\boldsymbol{h} = (h_{111}, h_{211}, \ldots, h_{sss})$ be a histogram vector, such that $h_{ijk} = |\{(u, v) \in E \mid \varphi(u, v) = i, \varphi(u) = j, \varphi(v) = k\}|$ for each $i, j, k \in \Sigma$. The vertex-edge label histogram kernel $K_{\mathrm{VEH}}(G, G')$ is defined as the linear kernel between $\boldsymbol{h}$ and $\boldsymbol{h}'$ for the respective histograms of $G$ and $G'$:
$$K_{\mathrm{VEH}}(G, G') = \langle \boldsymbol{h}, \boldsymbol{h}' \rangle = \sum_{i,j,k=1}^s h_{ijk} h_{ijk}'.$$

Notice that $K_{\mathrm{VEH}}(G, G') = K_{\mathrm{EH}}(G, G')$ if vertices are not labeled.

From the definition of the direct product of graphs, we can confirm the following relationships between histogram kernels and the random walk kernel.

**Lemma 1** *For a pair of graphs $G, G'$ and their direct product $G_\times$, we have*
$$K_{\mathrm{VH}}(G, G') = \frac{1}{\lambda_0} K_\times^0(G, G') = |V_\times|.$$

$$K_{\mathrm{VEH}}(G, G') = \frac{1}{\lambda_1} K_\times^1(G, G') - \frac{\lambda_0}{\lambda_1} K_\times^0(G, G') = \sum_{i,j=1}^{|V_\times|} [A_\times]_{ij}.$$

*Proof.* The first equation $K_{\mathrm{VH}}(G, G') = |V_\times|$ can be proven from the following:
$$K_{\mathrm{VH}}(G, G') = \sum_{v \in V} |\{ v' \in V' \mid \varphi(v) = \varphi'(v') \}| = |\{ (v, v') \in V \times V' \mid \varphi(v) = \varphi'(v') \}|$$
$$= |V_\times| = \frac{1}{\lambda_0} K_\times^0(G, G').$$

We can prove the second equation in a similar fashion:
$$K_{\mathrm{VEH}}(G, G') = 2 \sum_{(u,v) \in E} |\{ (u', v') \in E' \mid \varphi(u, v) = \varphi'(u', v'), \varphi(u) = \varphi'(u'), \varphi(v) = \varphi'(v') \}|$$
$$= 2 \left| \left\{ ((u, v), (u', v')) \in E \times E' \,\middle|\, \begin{array}{l} \varphi(u, v) = \varphi'(u', v'), \\ \varphi(u) = \varphi'(u'), \varphi(v) = \varphi'(v') \end{array} \right\} \right|$$
$$= 2|E_\times| = \sum_{i,j=1}^{|V_\times|} [A_\times]_{ij} = \frac{1}{\lambda_1} K_\times^1(G, G') - \frac{\lambda_0}{\lambda_1} K_\times^0(G, G'). \qquad \blacksquare$$

Finally, let us define a new kernel

$$K_{\mathrm{H}}(G, G') := K_{\mathrm{VH}}(G, G') + \lambda K_{\mathrm{VEH}}(G, G') \tag{1}$$

with a parameter $\lambda$. From Lemma 1, since $K_{\mathrm{H}}(G, G') = K_{\times}^{1}(G, G')$ holds if $\lambda_0 = 1$ and $\lambda_1 = \lambda$ in the one-step random walk kernel $K_{\times}^{1}$, we have the following relationship from Theorem 1.

**Corollary 3** *For a pair of graphs $G$ and $G'$, we have*

$$K_{\mathrm{H}}(G, G') \leq K_{\mathrm{GR}}(G, G') \leq K_{\mathrm{H}}(G, G') + \varepsilon,$$

*where $\varepsilon$ is given in Theorem 1.*

To summarize, our results show that if the parameter $\lambda$ of the geometric random walk kernel $K_{\mathrm{GR}}$ is small enough, random walks halt, and $K_{\mathrm{GR}}$ reduces to $K_{\mathrm{H}}$, which finally converges to $K_{\mathrm{VH}}$. This is based on vertex histograms only and completely ignores the topological structure of the graphs.

## 3 Experiments

We empirically examine the halting phenomenon of the geometric random walk kernel on popular real-world graph benchmark datasets and semi-simulated graph data.

### 3.1 Experimental Setup

**Environment.** We used Amazon Linux AMI release 2015.03 and ran all experiments on a single core of 2.5 GHz Intel Xeon CPU E5-2670 and 244 GB of memory. All kernels were implemented in `C++` with `Eigen` library and compiled with `gcc` 4.8.2.

**Datasets.** We collected five real-world graph classification benchmark datasets:[1] ENZYMES, NCI1, NCI109, MUTAG, and D&D, which are popular in the graph-classification literature [13, 14]. ENZYMES and D&D are proteins, and NCI1, NCI109, and MUTAG are chemical compounds. Statistics of these datasets are summarized in Table 1, in which we also show the maximum of maximum degrees of product graphs $\max_{G, G' \in \mathcal{G}} \Delta_{\times}$ for each dataset $\mathcal{G}$. We consistently used $\lambda_{\max} = (\max_{G, G' \in \mathcal{G}} \Delta_{\times})^{-1}$ as the upper bound of $\lambda$ in geometric random walk kernels, in which the gap was less than one order as the lower bound of $\lambda$. The average degree of the product graph, the lower bound of $\lambda$, were 18.17, 7.93, 5.60, 6.21, and 13.31 for ENZYMES, NCI1, NCI109, MUTAG, and DD, respectively.

**Kernels.** We employed the following graph kernels in our experiments: We used linear kernels on vertex label histograms $K_{\mathrm{VH}}$, edge label histograms $K_{\mathrm{EH}}$, vertex-edge label histograms $K_{\mathrm{VEH}}$, and the combination $K_{\mathrm{H}}$ introduced in Equation (1). We also included a Gaussian RBF kernel between vertex-edge label histograms, denoted as $K_{\mathrm{VEH,G}}$. From the family of random walk kernels, we used the geometric random walk kernel $K_{\mathrm{GR}}$ and the $k$-step random walk kernel $K_{\times}^{k}$. Only the number $k$ of steps were treated as a parameter in $K_{\times}^{k}$ and $\lambda_k$ was fixed to 1 for all $k$. We used fix-point iterations [14, Section 4.3] for efficient computation of $K_{\mathrm{GR}}$. Moreover, we employed the Weisfeiler-Lehman subtree kernel [13], denoted as $K_{\mathrm{WL}}$, as the state-of-the-art graph kernel, which has a parameter $h$ of the number of iterations.

### 3.2 Results on Real-World Datasets

We first compared the geometric random walk kernel $K_{\mathrm{GR}}$ to other kernels in graph classification. The classification accuracy of each graph kernel was examined by 10-fold cross validation with multiclass C-support vector classification (`libsvm`[2] was used), in which the parameter $C$ for C-SVC and a parameter (if one exists) of each kernel were chosen by internal 10-fold cross validation (CV) on only the training dataset. We repeated the whole experiment 10 times and reported average

`http://www.bsse.ethz.ch/mlcb/research/machine-learning/graph-kernels.html`
[2]`http://www.csie.ntu.edu.tw/~cjlin/libsvm/`

Table 1: Statistics of graph datasets, $|\Sigma_V|$ and $|\Sigma_E|$ denote the number of vertex and edge labels.

| Dataset | Size | #classes | avg.$|V|$ | avg.$|E|$ | max$|V|$ | max$|E|$ | $|\Sigma_V|$ | $|\Sigma_E|$ | max$\Delta_\times$ |
|---|---|---|---|---|---|---|---|---|---|
| ENZYMES | 600 | 6 | 32.63 | 62.14 | 126 | 149 | 3 | 1 | 65 |
| NCI1 | 4110 | 2 | 29.87 | 32.3 | 111 | 119 | 37 | 3 | 16 |
| NCI109 | 4127 | 2 | 29.68 | 32.13 | 111 | 119 | 38 | 3 | 17 |
| MUTAG | 188 | 2 | 17.93 | 19.79 | 28 | 33 | 7 | 11 | 10 |
| D&D | 1178 | 2 | 284.32 | 715.66 | 5748 | 14267 | 82 | 1 | 50 |

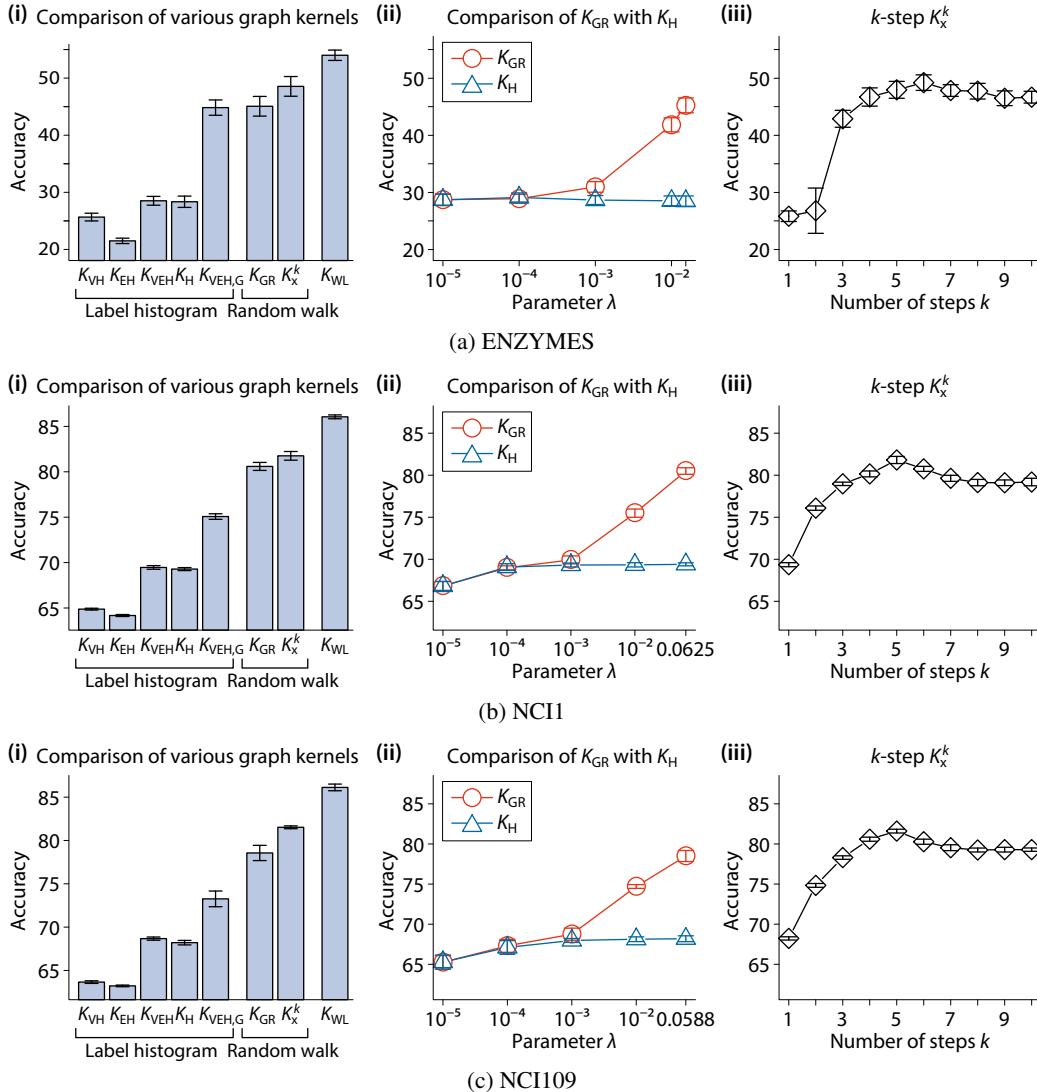

Figure 1: Classification accuracy on real-world datasets (Means $\pm$ SD).

classification accuracies with their standard errors. The list of parameters optimized by the internal CV is as follows: $C \in \{2^{-7}, 2^{-5}, \ldots, 2^5, 2^7\}$ for C-SVC, the width $\sigma \in \{10^{-2}, \ldots, 10^2\}$ in the RBF kernel $K_{\mathrm{VEH,G}}$, the number of steps $k \in \{1, \ldots, 10\}$ in $K_\times^k$, the number of iterations $h \in \{1, \ldots, 10\}$ in $K_{\mathrm{WL}}$, and $\lambda \in \{10^{-5}, \ldots, 10^{-2}, \lambda_{\max}\}$ in $K_{\mathrm{H}}$ and $K_{\mathrm{GR}}$, where $\lambda_{\max} = (\max_{G,G' \in \mathcal{G}} \Delta_\times)^{-1}$.

Results are summarized in the left column of Figure 1 for ENZYMES, NCI1, and NCI109. We present results on MUTAG and D&D in the Supplementary Notes, as different graph kernels do not give significantly different results (e.g., [13]). Overall, we could observe two trends. First, the Weisfeiler-Lehman subtree kernel $K_{\mathrm{WL}}$ was the most accurate, which confirms results in [13],

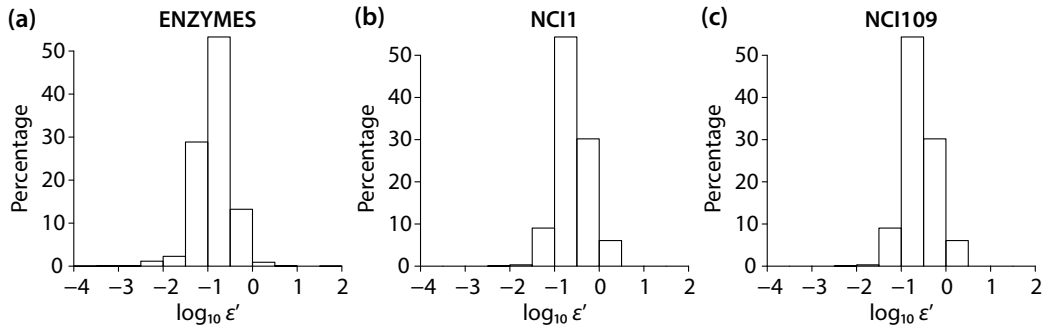

Figure 2: Distribution of $\log_{10} \varepsilon'$, where $\varepsilon'$ is defined in Corollary 1, in real-world datasets.

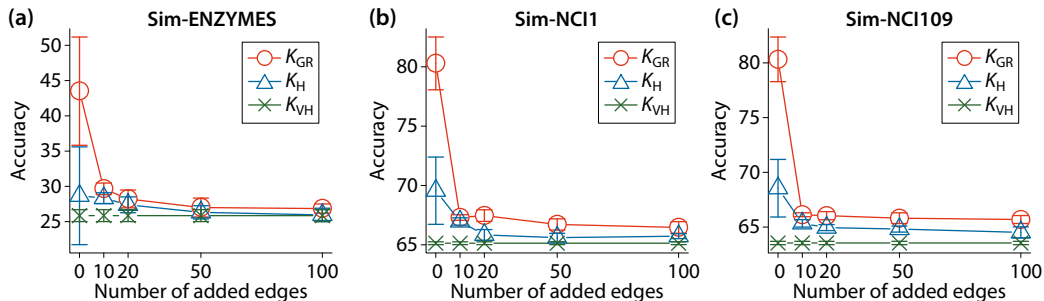

Figure 3: Classification accuracy on semi-simulated datasets (Means $\pm$ SD).

Second, the two random walk kernels $K_{\mathrm{GR}}$ and $K_\times^k$ show greater accuracy than naïve linear kernels on edge and vertex histograms, which indicates that halting is not occurring in these datasets. It is also noteworthy that employing a Gaussian RBF kernel on vertex-edge histograms leads to a clear improvement over linear kernels on all three datasets. On ENZYMES, the Gaussian kernel is even on par with the random walks in terms of accuracy.

To investigate the effect of halting in more detail, we show the accuracy of $K_{\mathrm{GR}}$ and $K_{\mathrm{H}}$ in the center column of Figure 1 for various choices of $\lambda$, from $10^{-5}$ to its upper bound. We can clearly see that halting occurs for small $\lambda$, which greatly affects the performance of $K_{\mathrm{GR}}$. More specifically, if it is chosen to be very small (smaller than $10^{-3}$ in our datasets), the accuracies are close to the naïve baseline $K_{\mathrm{H}}$ that ignores the topological structure of graphs. However, accuracies are much closer to that reached by the Weisfeiler-Lehman kernel if $\lambda$ is close to its theoretical maximum. Of course, the theoretical maximum of $\lambda$ depends on unseen test data in reality. Therefore, we often have to set $\lambda$ conservatively so that we can apply the trained model to any unseen graph data.

Moreover, we also investigated the accuracy of the random walk kernel as a function of the number of steps $k$ of the random walk kernel $K_\times^k$. Results are shown in the right column of Figure 1. In all datasets, accuracy improves with each step, up to four to five steps. The optimal number of steps in $K_\times^k$ and the maximum $\lambda$ give similar accuracy levels. We also confirmed Theorem 1 that conservative choices of $\lambda$ ($10^{-3}$ or less) give the same accuracy as a one-step random walk.

In addition, Figure 2 shows histograms of $\log_{10} \varepsilon'$, where $\varepsilon'$ is given in Corollary 1 for $\lambda = (\max \Delta_\times)^{-1}$ for all pairs of graphs in the respective datasets. The value $\varepsilon'$ can be viewed as the deviation of $K_{\mathrm{GR}}$ from $K_{\mathrm{H}}$ in percentages. Although $\varepsilon'$ is small on average (about 0.1 percent in ENZYMES and NCI datasets), we confirmed the existence of relatively large $\varepsilon'$ in the plot (more than 1 percent), which might cause the difference between $K_{\mathrm{GR}}$ and $K_{\mathrm{H}}$.

### 3.3 Results on Semi-Simulated Datasets

To empirically study halting, we generated semi-simulated graphs from our three benchmark datasets (ENZYMES, NCI1, and NCI109) and compared the three kernels $K_{\mathrm{GR}}$, $K_{\mathrm{H}}$, and $K_{\mathrm{VH}}$. In each dataset, we artificially generated denser graphs by randomly adding edges, in which the number of new edges per graph was determined from a normal distribution with the mean

$m \in \{10, 20, 50, 100\}$ and the distribution of edge labels was unchanged. Note that the accuracy of the vertex histogram kernel $K_{\mathrm{VH}}$ stays always the same, as we only added edges.

Results are plotted in Figure 3. There are two key observations. First, by adding new false edges to the graphs, the accuracy levels drop for both the random walk kernel and the histogram kernel. However, even after adding 100 new false edges per graph, they are both still better than a naïve classifier that assigns all graphs to the same class (accuracy of 16.6 percent on ENZYMES and approximately 50 percent on NCI1 and NCI109). Second, the geometric random walk kernel quickly approaches the accuracy level of $K_{\mathrm{H}}$ when new edges are added. This is a strong indicator that halting occurs. As graphs become denser, the upper bound for $\lambda$ gets smaller, and the accuracy of the geometric random walk kernel $K_{\mathrm{GR}}$ rapidly drops and converges to $K_{\mathrm{H}}$. This result confirms Corollary 3, which says that both $K_{\mathrm{GR}}$ and $K_{\mathrm{H}}$ converge to $K_{\mathrm{VH}}$ as $\lambda$ goes to 0.

# 4 Discussion

In this work, we show when and where the phenomenon of *halting* occurs in random walk kernels. *Halting* refers to the fact that similarity measures based on counting walks (of potentially infinite length) often downweight longer walks so much that the similarity score is completely dominated by walks of length 1, degenerating the random walk kernel to a simple kernel between edges and vertices. While it had been conjectured that this problem may arise in graph kernels [1], we provide the first theoretical proof and empirical demonstration of the occurrence and extent of halting in geometric random walk kernels.

We show that the difference between geometric random walk kernels and simple edge kernels depends on the maximum degree of the graphs being compared. With increasing maximum degree, the difference converges to zero. We empirically demonstrate on simulated graphs that the comparison of graphs with high maximum degrees suffers from halting. On real graph data from popular graph classification benchmark datasets, the maximum degree is so low that halting can be avoided if the decaying weight $\lambda$ is set close to its theoretical maximum. Still, if $\lambda$ is set conservatively to a low value to ensure convergence, halting can clearly be observed, even on unseen test graphs with unknown maximum degrees.

There is an interesting connection between halting and tottering [1, Section 2.1.5], a weakness of random walk kernels described more than a decade ago [11]. Tottering is the phenomenon that a walk of infinite length may go back and forth along the same edge, thereby creating an artificially inflated similarity score if two graphs share a common edge. Halting and tottering seem to be opposing effects. If halting occurs, the effect of tottering is reduced and vice versa. Halting downweights these tottering walks and counteracts the inflation of the similarity scores. An interesting point is that the strategies proposed to remove tottering from walk kernels did not lead to a clear improvement in classification accuracy [11], while we observed a strong negative effect of halting on the classification accuracy in our experiments (Section 3). This finding stresses the importance of studying halting.

Our theoretical and empirical results have important implications for future applications of random walk kernels. First, if the geometric random walk kernel is used on a graph dataset with known maximum degree, $\lambda$ should be close to the theoretical maximum. Second, simple baseline kernels based on vertex and edge label histograms should be employed to check empirically if the random walk kernel gives better accuracy results than these baselines. Third, particularly in datasets with high maximum degree, we advise using a fixed-length-$k$ random walk kernel rather than a geometric random walk kernel. Optimizing the length $k$ by cross validation on the training dataset led to competitive or superior results compared to the geometric random walk kernel in all of our experiments. Based on these results and the fact that by definition the fixed-length kernel does not suffer from halting, we recommend using the fixed-length random walk kernel as a comparison method in future studies on novel graph kernels.

**Acknowledgments.** This work was supported by JSPS KAKENHI Grant Number 26880013 (MS), the Alfried Krupp von Bohlen und Halbach-Stiftung (KB), the SNSF Starting Grant 'Significant Pattern Mining' (KB), and the Marie Curie Initial Training Network MLPM2012, Grant No. 316861 (KB).

## Footnotes

[1]The code and all datasets are available at:

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
