[Supplementary Material · camera_ready_supp.pdf]

# Supplementary Notes for
# Halting in Random Walk Kernels

**Mahito Sugiyama**
ISIR, Osaka University, Japan
JST, PRESTO
mahito@ar.sanken.osaka-u.ac.jp

**Karsten M. Borgwardt**
D-BSSE, ETH Zürich
Basel, Switzerland
karsten.borgwardt@bsse.ethz.ch

Figure S1: Accuracy results in real-world datasets (Means $\pm$ SD).

Figure S2: Distribution of $\log_{10} \varepsilon'$, where $\varepsilon'$ is defined in Corollary 1, in real-world datasets.