[Reviews · NeurIPS 2015]

Submitted by Assigned_Reviewer_1

-weak review-

The paper contains a theoretical proof of existence of the phenomenon of halting in geometric random kernels and an

extended experimental section that both supports the theoretical results and furthermore has surprising implications on kernel selection in practical contexts.

The paper is very well written and never displays any lack of clarity. The math is sound and only draws from rather basic concepts. The paper contains two main contributions: First, the authors provide a mathematical proof that halting can occur in geometric random walk kernels (which had previously been conjectured) by showing that certain kernel parameter choices lead the geometric random walk kernel to degrade into a simple topology-unaware kernel that is a superposition of a vertex label histogram kernel and a vertex-edge label histogram kernel. Secondly, the authors study the effect of halting empirically. For a variety of popular graph classification benchmark datasets, the authors observe halting to occur and are able to quantify its effects. The authors also arrive at a set of practical suggestions wrt reasonable use cases of geometric random walk kernels.

Major Comments The paper is very tidy and contains straight-forward reasoning. The main theoretical insight of the authors seems to have been the idea to study the (straight-forward) theoretical relationship between geometric random walk kernels and label histogram kernels. While the theoretical insights are rather trivial, the paper shines particularly in the experimental section: The authors very carefully compare and contrast the effects of kernel selection wrt naive graph kernels, k-step random walk kernels and geometric random walk kernels. In particular, they convincingly conclude that it is in general wise in practice to select a k-step random walk kernel instead of a geometric kernel, even though the latter may naively seem more powerful.

It would be nice to shed more light on why Gaussian RBF kernels on label histograms perform so favourably wrt sophisticated kernels.

Minor issues Thm 1 proof: "relationship(s) hold(s)" p. 8 last sentence "a (the) fixed-length random walk kernel "

The paper contains novel contributions, in particular a simple proof of a conjecture relevant to the ML community. The authors clearly show that their results effect and guide kernel selection in empirical contexts. Therefore, I propose to accept the paper for publication without further hesitation.

Summary: The paper contains novel contributions, in particular a simple proof of a conjecture relevant to the ML community. The authors clearly show that their results effect and guide kernel selection in empirical contexts. Therefore, I propose to accept the paper for publication without further hesitation.

Submitted by Assigned_Reviewer_2

This paper describes, with theoretical and empirical evidence, the occurrence of Halting in a particular family of graph kernels, based on random walks.

Halting is defined as the phenomenon where similarities between graphs are dominated by walks of shorter length.

This paper is overall well-written, showing aspects of both theoretical and empirical evidence.

The phenomemon of halting is a natural consequence of the geometric random walk kernel formulation, since the contributions of longer walks are lowered.

That said, the paper does not make a significant contribution but instead introduces a useful formal description of 'halting' and presents related experiments for validation.

To the authors:

+ Title gives the impression that Halting affects graph kernels in a broad sense, however, the paper describes how halting affects random walk graph kernels;

+ Table 1 shows column max|V| as real numbers (shouldn't those be integers?).

It is also strange to denote the number of vertex and edge labels using the same notation as the 'mapping function';

+ The analysis and discussion should include clear insights on how particular characteristics/statistics of each dataset (avg. number of edges, avg. number of vertices, etc.) have a connection (or effect) with the halting phenomenon and the behavior of random walk graph kernels.
Summary: This paper does not make a significant contribution but instead introduces a formal description of 'halting' in the context of random walk graph kernels and presents related experiments for analysis.

Submitted by Assigned_Reviewer_3

This paper investigates a known problem in random walk graph kernels, often referred as halting. Halting is a weakness in walk-based graph kernels, and happens due to the decaying factor \lambda down-weighting longer walks. Therefore, shorter (and mostly unsophisticated) walks which do not have distinguishing power dominates the similarity score; leading to an ineffective graph kernel. Authors theoretically analyze halting on random-walk kernels and perform experiments on benchmark graph kernel datasets and they share insights on how to fix halting problem in random walk kernels.

I think authors findings and analysis is useful and contributes to graph kernel literature. Authors extract useful insights such that how to set \lambda to solve halting based on their experiments, however the novelty of the paper seems limited for a conference like NIPS.

Moreover, authors compare against a few random-walk based kernels and edge/vertex based baselines, but only choose Weisfeiler-Lehman graph kernel as a comparative non-random-walk based kernel from the literature. It would be good to see at least the Shortest-Path graph kernel in the comparison list since it is known to outperform the random walk kernel (see reference [1] in the paper)

and avoids halting and tottering.

Also as a small suggestion; the connection between halting and tottering that the authors pointed out in Discussion section is previously mentioned by Karsten Bordwardt, 2007 (in his PhD thesis, page 55) therefore, it would be nice to cite this source in that paragraph.

Others: -- Unless I am mistaken, the authors do not discuss if (and how many times) they repeated the classification experiments for each dataset. Since they report error bars, I believe they run the experiments more than once, however, I am not able to see an explicit statement about it. -- While I think SEM refers to Standard Error of Mean, i think authors should have defined it somewhere for readers who are not familiar with the term. -- Typo: Line 243: Extra dot at the end of the sentence.
Summary: Authors investigate the problem of halting, a phenomenon in random-walk based graph kernels. Authors theoretically analyze halting on random-walk graph kernels, and share insights on how to fix halting based on their experiments on benchmark graph kernel datasets. While their findings and analysis contribute to graph kernel literature, the novelty of the paper seems limited.

Submitted by Assigned_Reviewer_4

The authors studied the halting problem is graph kernels. They demonstrate that the geometric series used for convergence in these kernels will down weight long paths of dense graphs. In this case, random walk kernels will converge to a trivial kernel on vertex and edges.

Pros: 1) The paper is well written and is self contained. 2) The mathematical proofs are clear and I enjoyed the effort to put the theorem into context following the proofs. 3) The empirical results are convincing.

Cons: 4) The authors briefly describe how halting can be avoided by comparing to existing methods during the Discussion but I would like more insight and in depth analysis on how to prevent halting. In particular, they mention that random walk of fixed length k will not suffer from halting. How about random walk of length 1 up to k or other approaches?

Questions / Points to address:

5) Line 96-102: the authors argue that setting \lambda to 1 / \Delta_{X, max} is sufficient to have \lambda < 1 / \mu_{X,max}. I am curious as to how tight this bound is. Given that the whole analysis is based on this inequality this warrant further analysis. 6) Line 161: generalisation of Theorem 1 to k-step random walk kernel should be provided as supplementary material. 7) Line 175: something is wrong with the syntax of the sentence. 8) Line 241-243: I strongly encourage the authors to provide their source code after acceptance. 9) Line 266: The values for the parameter C of SVM are not listed in those optimized by the internal CV.

The paper is a bit of a negative result but this should not negatively impact its chance for acceptation.
Summary: See pros and cons in comments to authors.

Author Feedback
Author rebuttal: We would like to thank all reviewers for their time and effort. We respond to major points raised by the reviewers. We will address all the other points in the final version.

TO ASSIGNED_REVIEWER_1:

> Title gives the impression that Halting affects graph kernels in a broad sense, however, the paper describes how halting affects random walk graph kernels;

A: We will revise our title according to your suggestion.

> Table 1 shows column max|V| as real numbers (shouldn't those be integers?).

A: Thank you for pointing this out. They are typo and numbers of avg.|E| and max|V| should be swapped except for MUTAG.

> It is also strange to denote the number of vertex and edge labels using the same notation as the 'mapping function';

A: We will clarify them. Please however note that they are mathematically correct as \phi(V) is the set of vertex labels.

> The analysis and discussion should include clear insights on how particular characteristics/statistics of each dataset (avg. number of edges, avg. number of vertices, etc.) have a connection (or effect) with the halting phenomenon and the behavior of random walk graph kernels.

A: As we have shown in Theorem 1, the maximum degree of a product graph is crucial for the halting phenomenon because its inverse is the upper bound of the parameter \lambda. Thus the number of edges is relevant as the degree becomes high in a dense graph. We have empirically confirmed this on semi-simulated data (Figure 3) by inserting edges. We will revise our discussion to clarify these points.

TO ASSIGNED_REVIEWER_2:

> Moreover, authors compare against a few random-walk based kernels and edge/vertex based baselines, but only choose Weisfeiler-Lehman graph kernel as a comparative non-random-walk based kernel from the literature. It would be good to see at least the Shortest-Path graph kernel in the comparison list since it is known to outperform the random walk kernel (see reference [1] in the paper) and avoids halting and tottering.

A: The aim of our experiments is 1) to analyze the halting phenomenon of random walk-based kernels, and 2) to confirm the performance of the state-of-the-art method, that is, the Weisfeiler-Lehman graph kernel. Since the shortest-path graph kernel is neither a random walk-based kernel nor the state-of-the-art, we believe that it is not very relevant in our paper.

> Also as a small suggestion; the connection between halting and tottering that the authors pointed out in Discussion section is previously mentioned by Karsten Borgwardt, 2007 (in his PhD thesis, page 55) therefore, it would be nice to cite this source in that paragraph.

A: Thank you for pointing out that we skipped the citation. We will also cite it in the paragraph (3rd paragraph in Section 4).

> Unless I am mistaken, the authors do not discuss if (and how many times) they repeated the classification experiments for each dataset.

A: We performed 10-fold cross validation (see line 263), thereby each mean and standard error of the mean (SEM, we will clarify it) are computed from 10 accuracies. We will repeat experiments 10 times for the final version.

TO ASSIGNED_REVIEWER_3:

> 4) The authors briefly describe how halting can be avoided by comparing to existing methods during the Discussion but I would like more insight and in depth analysis on how to prevent halting. In particular, they mention that random walk of fixed length k will not suffer from halting. How about random walk of length 1 up to k or other approaches?

A: Thank you for the suggestion. Since the decaying weight \lambda causes halting, we can avoid halting by using a kernel that does not require decaying \lambda. We have examined and recommended the k-step random walk kernel K^k_x as a representative. Please note that walks of length up to k are already taken into account in the k-step random walk kernel K^k_x (see line 80).

> 5) Line 96-102: the authors argue that setting \lambda to 1 / \Delta_{X, max} is sufficient to have \lambda < 1 / \mu_{X,max}. I am curious as to how tight this bound is.

A: Although we could not check \mu_{X,max} for all datasets due to high computational cost, we have checked the average degree, the lower bound of it (see line 100). They are 18.17, 7.93, 5.60, 6.21, and 13.31 for ENZYMES, NCI1, NCI109, MUTAG, and DD, respectively. As you can see, the difference is less than one order in all datasets. We will include these numbers.

> 6) Line 161: generalisation of Theorem 1 to k-step random walk kernel should be provided as supplementary material.

A: Corollary 2 (line 162) is the generalization of Theorem 1.

> 9) Line 266: The values for the parameter C of SVM are not listed in those optimized by the internal CV.

A: Thank you for pointing this out. C is from {2^(-7), 2^(-5), 2^(-3), 2^(-1), 2^1, 2^3, 2^5, 2^7}. We will add it.